

# Categorization of Alzheimer's disease stages using deep learning approaches with McNemar's test

Begüm Şener[1], Koray Acici[2] and Emre Sümer[1]

[1] Department of Computer Engineering, Başkent University, Ankara, Başkent University, Ankara, Turkey
[2] Department of Artificial Intelligence and Data Engineering, Ankara University, Ankara, University, Ankara, Turkey

## ABSTRACT

Early diagnosis is crucial in Alzheimer's disease both clinically and for preventing the rapid progression of the disease. Early diagnosis with awareness studies of the disease is of great importance in terms of controlling the disease at an early stage. Additionally, early detection can reduce treatment costs associated with the disease. A study has been carried out on this subject to have the great importance of detecting Alzheimer's disease at a mild stage and being able to grade the disease correctly. This study's dataset consisting of MRI images from the Alzheimer's Disease Neuroimaging Initiative (ADNI) was split into training and testing sets, and deep learning-based approaches were used to obtain results. The dataset consists of three classes: Alzheimer's disease (AD), Cognitive Normal (CN), and Mild Cognitive Impairment (MCI). The achieved results showed an accuracy of 98.94% for CN *vs* AD in the one *vs* one (1 *vs* 1) classification with the EfficientNetB0 model and 99.58% for AD *vs* CNMCI in the one *vs* All (1 *vs* All) classification with AlexNet model. In addition, in the study, an accuracy of 98.42% was obtained with the EfficientNet121 model in MCI *vs* CN classification. These results indicate the significant potential for mild stage Alzheimer's disease detection of Alzheimer's disease. Early detection of the disease in the mild stage is a critical factor in preventing the progression of Alzheimer's disease. In addition, a variant of the non-parametric statistical McNemar's Test was applied to determine the statistical significance of the results obtained in the study. Statistical significance of 1 *vs* 1 and 1 *vs* all classifications were obtained for EfficientNetB0, DenseNet, and AlexNet models.

## INTRODUCTION

Alzheimer's disease (AD), a common type of dementia, is a neurological disease that destroys brain cells, reduces cognitive function, and occurs as a result of the accumulation of toxic proteins in some parts of the brain. Symptoms of this disease appear gradually with age. The disease, which begins with simple forgetfulness in the initial stage, can progress to symptoms such as the patient forgetting events in their own recent past and losing the ability to recognize their immediate environment and family members as time progresses.

Corresponding author
Begüm Şener,
begume@baskent.edu.tr

In the later stages of the disease, it may be challenging to meet the patient's basic needs and provide care (*Cummings & Cole, 2002*).

Detecting AD at an early stage is both clinically crucial and valuable for preventing the rapid progression of the disease. Studies focusing on early diagnosis based on disease awareness are important for controlling the disease at an early stage. While this situation improves the quality of life both for the patients and their relatives, it also reduces the treatment costs of the disease. Several imaging techniques have been associated with diagnosing AD (*Colligris et al., 2018*). AD progresses through three stages (*Hart et al., 2003*). The early stage typically lasts 2 to 4 years. During this stage, individuals experience frequent short-term memory problems, repetitions in speech and questions, mild difficulties in self-expression, difficulties in writing and using objects, the development of depression, personality changes, the inability to learn new skills, denial of the disease, and irritability. During the middle stage, the patient might experience progressive memory impairments, disturbances in orientation, difficulties in establishing cause–effect relationships, sleep disorders, and an inability to acquire new information. In the advanced stage, the patient might experience confusion between the past and present, severe impairment in communication, falls, bed dependency, swallowing problems, pronounced psychiatric symptoms, and being entirely in need of care.

The radiological approaches used for the diagnosis of AD include computed tomography (CT), magnetic resonance imaging (MRI), functional MRI (fMRI), positron emission tomography (PET), and single-photon emission computed tomography (SPECT). These are standard procedures for diagnosis. In MRI, three separate images of the same region are available, each with different tissue contrast: T1-weighted, T2-weighted, and proton-weighted images (*Buxton et al., 1987*). Water appears black on T1-weighted images, white on T2-weighted images, and gray on proton-weighted images. Because there were few uses of proton-weighted images, these images were removed from standard reviews.

The inspiration for studying this problem was found during the examination of other studies in the literature. These studies found that advanced-stage AD can be detected relatively easily, whereas mild-stage AD is more difficult to detect. The necessity and significance of detecting AD in the mild stage is the reason for conducting this study.

The main aims of the study are as follows:

- Demonstrate the performance of deep learning models in the diagnosis of AD during the middle stage of the disease.
- Show the effects of classifying classes in models as one-*versus*-one or one-*versus*-all.
- Overcome the difficulties in identifying patients with cognitive normal (CN) and mild cognitive impairment (MCI) and detect AD at the mild stage before it progresses.

The manuscript is structured as follows: First, we briefly describe AD. Afterward, deep learning techniques and the literature on AD are presented as an overview. After defining the dataset, the methodology is presented. The methodology section provides definitions

and evaluation metrics of the models used. Finally, we discuss the results obtained and present our conclusions, including recommendations for future work.

## RELATED WORK

There are numerous studies in the literature on the diagnosis of AD. A summary of the literature review on AD is presented in the following section.

*Mora-Rubio et al. (2023)* presented a deep learning-based approach to classifying MRI scans at different stages of AD using a set of images compiled from the Alzheimer's Disease Neuroimaging Initiative (ADNI) and Open Access Series of Imaging Studies (OASIS). They used data augmentation operations such as preprocessing, rotation, translation, and zooming using FreeSurfer. They obtained results using state-of-the-art convolutional neural networks (CNNs) such as EfficientNet and DenseNet. The best results obtained were 89% for CN *vs* AD, 80% for CN *vs* late MCI (LMCI), 66% for CN *vs* MCI, and 67% for CN *vs* early MCI (EMCI).

*Shanmugam et al. (2022)* utilized the ADNI dataset to work on five classes: CN, EMCI, MCI, LMCI, and AD. Using a total of 7,800 images, they divided the image data of the five classes into 60% for training, 20% for testing, and 20% for validation. The results were obtained over 100 epochs. The images were adjusted to the required dimensions for processing through the respective models. Using transfer learning, the study proposed three different deep learning models, namely AlexNet, ResNet-18, and GoogLeNet. According to the results, the AlexNet model achieved an accuracy of 97.34% for CN, 97.51% for EMCI, 95.19% for LMCI, 96.82% for MCI, and 94.08% for AD. The ResNet-18 model achieved an accuracy of 98.88% for CN, 99.14% for EMCI, 98.88% for LMCI, 98.71% for MCI, and 97.51% for AD. The GoogLeNet model achieved an accuracy of 97.17% for CN, 98.28% for EMCI, 97.60% for LMCI, 98.37% for MCI, and 96.39% for AD. The authors are planning future work focused on designing deep learning networks specifically for classifying AD cases.

In the study by *Mehmood et al. (2021)*, T1-weighted MRI images from 300 AD patients in the ADNI dataset were used. Their proposed transfer learning model divided the layers into two groups, gradually training some layers while freezing the remaining ones. In the recommended neural network models, Group A had three max-pooling convolutional layers, while Group B had four max-pooling convolutional layers, which were frozen. They used six binary classifications and evaluated the performance of their proposed transfer learning model: CN *vs* EMCI, CN *vs* AD, CN *vs* LCMI, EMCI *vs* LMCI, EMCI *vs* AD, and LMCI *vs* AD. They evaluated the models with and without data augmentation. The data were divided into 80% for training and 20% for testing. They also used some of the 20% test MRI images for validation. For Group A, the highest accuracy without data augmentation was achieved in comparing CN *vs* AD, with an accuracy of 93.83%. With data augmentation, the highest accuracy in the CN *vs* AD comparison reached 95.38%. For Group B, the highest accuracy without data augmentation was achieved in comparing CN *vs* AD, with an accuracy of 95.33%. With data augmentation, the highest accuracy in the CN *vs* AD comparison reached 98.73%.

*Mohammadjafari et al. (2021)* experimented with CNN methods using the OASIS and ADNI AD datasets. They utilized the ADNI-1 dataset, which consists of 95 MRI scans of AD patients and 113 MRI scans of healthy individuals (CN). The images were resized to 224 × 224 and converted to red, green, and blue (RGB) bands to fit the training configurations models. They used five-fold cross-validation to validate the models. They obtained average accuracy for the baseline model, transfer learning with the baseline model, and transfer learning with the ProtoPNet model. For the baseline model, they achieved an accuracy of 75.76% for VGG-16, 77.06% for ResNet50, and 73.23% for DenseNet121. When performing transfer learning with the ProtoPNet model, the observed accuracies were 71.70% for VGG-16, 90.30% for ResNet50, and 91.02% for DenseNet121.

*Sethi et al. (2022)* utilized the ADNI dataset to conduct a study on the MRI images of 50 patients for each class (AD, CN, and MCI). They used 80% of the dataset for training and 20% for model testing. They used 23,232 images for training and 6,468 images for testing. They achieved results by combining CNN with support vector machines (SVM) as a hybrid model. When only the CNN model was used, the highest accuracy rate was 82.32% for CN *vs* AD, and the test accuracy rate was 85.1%. When the hybrid model of CNN and SVM was used, the highest accuracy rate was again 89.4% for CN *vs* AD, and the test accuracy rate was 88%.

*Naz, Ashraf & Zaib (2021)* studied T1-weighted MRI images using the ADNI dataset. They evaluated 11 pretrained CNN models. They extracted features from FC CNN layers and studied the MCI-AD, AD-CN, and MCI-CN classes. The dataset contained 95 AD, 95 CN, and 146 MCI. Each patient had a different number of scans, ranging from a minimum of three to a maximum of 15. Digital Imaging and Communications in Medicine (DICOM) images were converted to Joint Photographic Experts Group (JPEG) format. By augmenting the dataset, they obtained 37,590 images after augmentation, whereas the original dataset had 3,925 images. They divided the dataset into 80% for training, 10% for testing, and 10% for validation. Using the frozen features extracted from the augmented dataset and the VGG-19 model, they achieved the highest accuracy rate of 99.27% for MCI *vs* AD. The VGG-16 model obtained an accuracy rate of 98.89% for CN *vs* AD and 97.06% for CN *vs* MCI. Using the AlexNet model, they obtained an accuracy rate of 91.38% for CN *vs* AD.

In their study, *Farooq et al. (2017)* used the ADNI dataset of 149 AD patients. They divided the dataset into 33 AD, 22 LMCI, 49 MCI, and 45 CN as 75% training and 25% testing and used 10% of the training set for validation. To eliminate the imbalance in each class in the dataset, they used 38,024 images by increasing the number of images of the missing classes. They obtained an average accuracy of 98.88% for the AlexNet model, an average accuracy of 98.01% for ResNet-18, and an average accuracy of 98.14% for ResNet-152.

The study by *Savaş (2022)* used T1-weighted and sagittal images in the ADNI-1 dataset. There were 382 patients in the dataset, of which 223 were male and 159 were female. The dataset was split into 90% for training and 10% for testing, and 10% of the 90% training portion was used for validation. Images were resized to 224 × 224. The dataset has three classes: 135 CN images, 148 MCI images, and 99 AD images. Deep neural network

architectures created using the CNN algorithm were used to analyze the data employed in the study. Models other than the ResNet model were set to run for 250 epochs because the learning increment in the other models continued. In the test results of the model, EfficientNetB0 provided the best result, with 92.98%, followed by EfficientNetB1, with 91.91%. The lowest accuracy rate was 77.40% with the Xception model. The AD sensitivity value for EfficientNetB0 was 94.34%, the sensitivity value for MCI was 94.25%, and the sensitivity value for CN was 86.17%. The specificity value, on the other hand, was 96.96% for CN at the highest rate.

*Li, Cheng & Liu (2017)* conducted a study that utilized T1-weighted sagittal MRI images from the ADNI dataset. There are 427 AD patients in the dataset, of which 199 have AD and 229 are healthy (CN). They used five-fold cross-validation to evaluate classification performance and to train and test models. First, they obtained the results for the individual models they proposed, and then they obtained results by combining these models. They used MRI images sized $69 \times 59 \times 57$ for CNN_S3, $102 \times 88 \times 85$ for CAE_S2, $69 \times 59 \times 57$ for CAE_S3, and $54 \times 44 \times 43$ for CAE_S4. They achieved 84.12% accuracy for the CNN_S3 model, 82.24% for the CAE_S2 model, 81.19% for the CAE_S3 model, 76.17% for the CAE_S4 model, and 88.31% accuracy with all of the models combined.

In their study, *Khan et al. (2022)* used the ADNI dataset divided into 70% for training and 30% for testing. The dataset contained 2,127 images that were axial and T1- and T2-weighted. Their data included 612 AD, 538 MCI, and 975 CN classes. To ensure the highest pixel quality, the images were resized to dimensions of $512 \times 512$. Afterward, the images were standardized and normalized. They obtained results using extreme gradient boosting (XGB), decision trees (DTs), and decision support machines. They achieved an accuracy of 89.77% in the XGB + DT + SVM hybrid model. Subsequently, all models' efficiency was optimized using grid-based tuning, and they saw a significant improvement in the results obtained after this process. The best average accuracy rate was 95.75% due to the optimized parameters. They achieved an accuracy rate of 96.12% for the AD class, 95% for the CN class, and 96.15% for the MCI class.

*Mohi ud din dar et al. (2023)* studied the ADNI dataset using T2-weighted MRI images. The dataset included data from 300 AD patients divided into five classes: CN, MCI, EMCI, LMCI, and AD. There were 1,101 images in the LMCI class, 493 in the AD class, 204 in the EMCI class, 61 in the LMCI class, 198 in the MCI class, and 493 in the CN class. They resized all the images to $224 \times 224$ and processed RGB images using three channels. Because of the dataset's instability, the insufficient data were multiplied by the data augmentation method in the form of 580 MRI images for each class. The dataset was balanced, and a total of 2,900 images were processed. In the enhancement method, some techniques, such as horizontal flip and five-degree rotation, were used in the images. They divided the dataset into 80% for training, 10% for testing, and 10% for validation. The images were normalized during preprocessing. Using the CNN architecture, the researchers achieved an average accuracy rate of 96.22% after training for 100 epochs.

*Islam & Zhang (2017)* used the OASIS dataset in their study. They performed classification using T1-weighted MRI scans of the OASIS dataset, which comprised four classes and 416 subjects. They used the Inception-v4 model to create a framework with

**Table 1 The summary of the studies.**

| Reference | Year | Dataset | Models | Classes | Accuracy % |
|---|---|---|---|---|---|
| *Shanmugam et al. (2022)* | 2022 | ADNI | AlexNet | CN | 97.34% |
| | | | | EMCI | 97.51% |
| | | | | LMCI | 95.19% |
| | | | | MCI | 96.82% |
| | | | | AD | 94.08% |
| | | | ResNet-18 | CN | 98.88% |
| | | | | EMCI | 99.14% |
| | | | | LMCI | 98.88% |
| | | | | MCI | 98.71% |
| | | | | AD | 97.51% |
| | | | GoogleNet | CN | 97.17% |
| | | | | EMCI | 98.28% |
| | | | | LMCI | 97.60% |
| | | | | MCI | 98.37 |
| | | | | AD | 96.39% |
| *Mehmood et al. (2021)* | 2021 | ADNI | CNN | CN *vs* AD (Group A) | 95.38% |
| | | | | CN *vs* AD (Group B) | 98.73% |
| *Mohammadjafari et al. (2021)* | 2021 | ADNI-1 | VGG-16 | AD, CN | 88.50% |
| | | | ResNet50 | | 83.88% |
| | | | DenseNet121 | | 94.75% |
| *Sethi et al. (2022)* | 2022 | ADNI | CNN | CN *vs* AD | 82.32% |
| | | | CNN+SVM | | 89.40% |
| *Naz, Ashraf & Zaib (2021)* | 2021 | ADNI | VGG-19 | MCI *vs* AD | 99.27% |
| | | | VGG-16 | CN *vs* AD | 98.89% |
| | | | AlexNet | CN *vs* AD | 91.38% |
| | | | VGG-16 | MCI *vs* CN | 97.06% |
| *Farooq et al. (2017)* | 2017 | ADNI | AlexNet | AD, LMCI, MCI, CN | 98.88% |
| | | | ResNet-18 | | 98.01% |
| | | | ResNet-152 | | 98.14% |
| *Savaş (2022)* | 2022 | ADNI | EfficientNetB0 | CN, MCI, AD | 92.98% |
| | | | EfficientNetB1 | | 91.91% |
| *Li, Cheng & Liu (2017)* | 2017 | ADNI | CNN_S3 | CN, AD | 84.12% |
| | | | CAE_S2 | | 82.24% |
| | | | CAE_S3 | | 81.19% |
| | | | CAE_S4 | | 76.17% |
| | | | Hybrid | | 88.31% |
| *Khan et al. (2022)* | 2022 | ADNI | XGB + DT + SVM | CN, MCI, AD | 95.75% |
| *Mohi ud din dar et al. (2023)* | 2023 | ADNI | CNN | CN, LMCI, EMCI, MCI, AD | 96.22% |
| *Mora-Rubio et al. (2023)* | 2023 | ADNI, OASIS | DenseNet | CN *vs* MCI | 66.41% |
| | | | EfficientNet | | |
| | | | | CN *vs* AD | 89.02% |
| | | | VIT | CN *vs* LMCI | 80.56% |
| | | | Siamese | CN *vs* EMCI | 67.19% |

suggestions. The five-fold cross-validation method was preferred while running the model. The results were obtained by running the model for both five and 10 epochs; the researchers obtained an accuracy rate of 71.25% after training for five epochs and 73.75% after training for 10 epochs.

*Mamun et al. (2022)* obtained the dataset for their study online *via* Kaggle. It comprised 6,219 MRI images with four classes. After the dataset was preprocessed by resizing images, removing noises, segmenting images, and performing smoothing operations, the dataset was trained using the holdout cross-validation method. They obtained results using a proposed CNN architecture, and the highest accuracy rate they achieved was 97.60%. In addition, other metrics, such as AUC, recall, and loss values, were given along with the statistical analysis comments.

*Sekhar & Jagadev (2023)* used an AD MRI preprocessed dataset in their study. The dataset comprised 6400 MRI images with four classes. The dataset was divided into train, test, and validation portions. The researchers found that the advantage of end-to-end learning was that it could create an effective visual explanation of the logic behind the classification results obtained using CNN, which will help medical experts understand the impact of CNN and find new biomarkers. The best result they obtained was 98.5% with the EfficientNet model.

A summary of the literature is provided in Table 1.

To summarize, the studies reviewed in this article present the most commonly proposed models and innovations in classifying AD using deep learning techniques. In addition to the one-*versus*-one classification, we also present the results obtained from the one-*versus*-all classification, an approach that has been less common in AD research. In the next sections, we present a study performed using a different, more recent dataset and explore the potential changes in model performance.

## DATASET

The ADNI dataset, which was obtained from the ADNI database (accessible *via* the website http://adni.loni.usc.edu), was employed in this study. Launched in 2016, the ADNI-3 dataset aims to comprehensively identify relationships between clinical, genetic, imaging, cognitive, and biochemical biomarker features across the entire spectrum of AD. ADNI-3 also includes brain scans that detect tau protein tangles (tau PET), a crucial disease indicator. The ADNI datasets were analyzed for AD detection. Because we found that the most recent data were in the ADNI-3 dataset, our analysis studies were started using the ADNI-3 dataset.

In this study, T1-weighted structural MRI data and axial brain slices from the ADNI-3 dataset were used for 515 AD patients, including 40 AD, 140 MCI, and 335 CN patients. Multiple scans of each subject were taken at different times, and each subject had a different number of scans. There were 259 female and 256 male individuals in the dataset. The dataset included data that were last released in November 2022. Because of the unbalanced class distributions in the dataset, it was arranged to balance the number of brain scans in each class. The Python programming language was used to conduct these analyses.

**Table 2 Classes and total scan numbers of these classes are given.**

| Class | Subjects | Total scans |
|---|---|---|
| AD | 40 | 4,230 |
| MCI | 140 | 5,961 |
| CN | 335 | 6,575 |

Individuals with AD had received an AD diagnosis before the beginning of the study period. CN subjects were in good health from the beginning of the study period and maintained good health throughout the study. MCI subjects had begun to show symptoms of AD by the beginning of the study period but had not yet fully progressed to AD and were not as healthy as the CN subjects. Table 2 presents the classes, number of subjects, and total scan numbers of the classes.

The image data format of the images in the dataset was Neuroimaging Informatics Technology Initiative (NIFTI), which has a file extension of .nii. These image files were converted to portable network graphic (PNG) images. During data preprocessing, each input MRI image in our CNN model was resized to 224 × 224 before it was fed into the model because the model architectures used 224 × 224 input images.

## METHODOLOGY

### Convolutional neural network

The CNN model's architecture consists of five convolution blocks, pooling layers, and a fully connected (FC) layer. FC layers are used to calculate the output of each input MRI image. In the FC layer, the image that passes through the convolutional and pooling layers several times and is in the form of a matrix is transformed into a flat vector.

The labeling process was initiated after data collection, image resizing, and the conversion of NIFTI images to PNG format. The label classes from the images and the comma-separated values (CSV) file were mapped to each other. In the next step, the dataset was divided into 70% for training and 30% for testing. A total of 10% of the 70% training portion was selected as the validation set and was used during the data evaluation process.

The EfficientNetB0, DenseNet121, and AlexNet CNN models were applied to MRI images using (TensorFlow, 2023) and Keras (Costa, 2023) applications. When we selected the models, the importance of each model was taken into consideration. AlexNet and DenseNet121 models have shown that CNNs have revolutionized image processing. The AlexNet model has a relatively simple design, which makes it relatively easy to train and implement. This is important for applications in which time and cost are essential, such as AD classification. Because of its impressive performance, simplicity, and generality, the AlexNet model was chosen; we believed that it would provide significant advantages for this application. Because the EfficientNetB0 model is a CNN model that shows impressive results in terms of scalability and performance, it can achieve high accuracy and performance in classification tasks using fewer parameters and less computational power.

Because of these important and powerful features, we preferred to use the EfficientNetB0 model over the other models.

While running the models, the relationship between the values obtained in each epoch was observed and care was taken to avoid overfitting. The optimization of CNN models using the dropout process is an important step to improve the performance of the models by preventing overfitting. Thus, the dropout process was performed for the three models used. The dropout process also increases the learning capacity of the models, making them more robust. Pooling was also conducted for each model, which helped the model train faster and reduced the computational power required by shrinking the images. We also investigated the rationale behind avoiding traditional machine learning models such as XGBoost or LightGBM. Although these models are effective in many situations, difficulties in terms of memory utilization and computational power can be encountered when working with large datasets. The symptoms of AD can be complex and multifaceted, and the reliability of models for medical diagnostics is therefore critical. Models such as LightGBM or XGBoost may have limitations in accurately capturing this complexity.

## Convolutional layer

The convolutional layer is an artificial neural network layer that is a fundamental component of deep learning and feature extraction. These layers are structures in which successful results can be obtained, especially in the use cases of image processing and pattern recognition. The convolution process is applied to the input data. It can create multiple feature maps using many different filters. Convolutional layers are often used with activation functions (rectified linear unit (ReLU)) and sequentially applied to convolution, pooling, and subsampling layers (*Albawi, Mohammed & Al-Zawi, 2017*).

## Pooling layer

Pooling layers are a type of layer used in CNN models. Pooling layers can be used to reduce data size; they can also merge feature maps and protect critical information. The two most commonly used pooling methods are maximum pooling and average pooling. Maximum pooling creates a summary of the region by selecting the highest feature value within each region to preserve its most important features and maintain its originality. Average pooling summarizes the region by averaging the feature values within each region. This method reduces noise and provides a smoother summary of the features. Pooling layers are usually used after many convolutional layers, thus reducing the size of feature maps and summarizing the output data. Pooling layers help deep learning models learn more general and high-level features and increase the network's generalization ability (*Sun et al., 2017*).

## Fully connected layer

The FC layer is an artificial neural network layer, also known as a densely connected layer. It receives the outputs of all units (neurons) in the previous layer and connects each output with its neurons. It can be the last layer of a neural network model (to provide an output such as classification or regression), or it can be used in the middle layers of the model. FC layers are widely used in deep learning models. These usually contain a large number of

neurons and increase the learning capacity of the model. These layers are often combined with activation functions (ReLU, sigmoid, *etc.*) to add nonlinearity to the inputs. In summary, the FC layer is a neural network layer that connects the input vector to all output units, increases the learning capacity of the neural network, and helps the model learn complex relationships (*Basha et al., 2020*).

## Softmax classification layer

The Softmax classification layer is an output layer used to address classification problems in artificial neural network models. This layer transforms a model's inputs into an output vector representing probabilities that can be assigned to classes. Using the Softmax function, this layer transforms the network's outputs into class probabilities. The input vector values reaching the layer are initially processed by the Softmax function. The Softmax function converts the input values into probability values to calculate the value of each output unit. This transformation provides a probability distribution in which the sum of all output units equals 1 (*Maharjan et al., 2020*).

## EfficientNetB0

EfficientNetB0 is a widely used image classification model in deep learning. EfficientNet is a family of models developed by Google Brain and optimized for scalability, efficiency, and performance. B0 represents the lowest value of the scaling coefficient of the model. EfficientNetB0 is an optimized model designed for high-performance image classification problems using the CNN's powerful features. The model's features and performance have been carefully designed with scaling strategies such as weight sharing, depth scaling, and width scaling. Because of this, although it is a lighter model, it can provide similar or better performance than other larger and more complex models. It is a model that is generally preferred for use on small and medium-sized datasets. However, higher-scale EfficientNet (B1, B2, B3, …) models are also available for larger, more complex datasets and more demanding tasks (*Marques, Agarwal & De la Torre Díez, 2020*).

## DenseNet121

DenseNet is a family of models based on the concept of dense connections. DenseNet121 is a widely used image classification deep learning model that is a member of this model family and consists of 121 layers. The model's name refers to the number of layers in which dense connections are used. The DenseNet structure, unlike traditional CNNs, uses dense connections in each layer in which the outputs of all previous layers are used as inputs. These dense connections facilitate the flow of information by combining the previous layer's outputs with the next layer's inputs. In this way, each layer can access the outputs of all previous layers. Dense connections improve the effectiveness and efficiency of information and mitigate the gradient loss problem regardless of the depth of the network. DenseNet121 improves performance by combining the power of CNNs with dense connections. This model is generally preferred for use on medium-sized datasets. Trained on the ImageNet dataset, DenseNet121 is tailored for tasks such as image classification.

This model has been used successfully in many deep learning projects and image analysis applications (*Solano-Rojas, Villalón-Fonseca & Marín-Raventós, 2020*).

### AlexNet

AlexNet represents a significant milestone within deep learning and CNNs and is an artificial neural network model of paramount importance. It was developed in 2012 by Alex Krizhevsky, Geoffrey Hinton, and Ilya Sutskever and features a much deeper structure than those of other models from that period. It contains eight layers, five of which are convolutional layers and two of which are FC consecutive layers. AlexNet pioneered the widespread use of CNNs and made a significant impact on the field of deep learning (*Omonigho et al., 2020*).

### One-*versus*-one classification

One-*versus*-one is a method that treats multiclass classification problems as pairwise comparisons between two classes. A separate classifier is created for both classes, and a pair of classifiers is created for each class combination. For example, if you have a total of $N$ classes, the one-*versus*-one method creates a total of $(N \times (N - 1))/2$ classifiers. The one-*versus*-one method increases the ease and speed of solving each binary classification problem. It also allows for the use of more common binary classification algorithms as multiclass classification algorithms, transforming multiclass classification into binary classification (*Lingras & Butz, 2007*).

### One-*versus*-all classification

One-*versus*-all is a method that treats multiclass classification problems as one class, which distinguishes each class from the others, as well as all remaining classes. This method aims to solve multiclass classification problems by transforming them into binary classification problems. In the one-*versus*-all method, a separate classifier (for example, a binary classifier or binary classification model) is created for each class. When a separate classifier is created for each class, this classifier is trained to distinguish that class from other classes. In contrast, all remaining classes are combined to form a single "other" class. The one-*versus*-all method requires as many classifiers as there are classes. The advantage is that binary classification algorithms can be used, and each class can be learned separately. This provides a more flexible solution when class labels are unstable or the complexity is different between classes (*Lingras & Butz, 2007*).

### Performance evaluation metrics

Various metrics are used to evaluate the performance of models. Accuracy provides an overview of the prediction quality and indicates the proportion of correctly classified samples. On the other hand, precision indicates the proportion of true positive predictions among the values we predicted as positive. Sensitivity (recall) is a metric that demonstrates the proportion of positive instances correctly predicted among the actual positive instances. The F1 score is a performance metric that aims to strike a balance between precision and sensitivity and is calculated by using the harmonic mean. The area under the curve (AUC) represents the area under the receiver operating characteristic (ROC) curve.

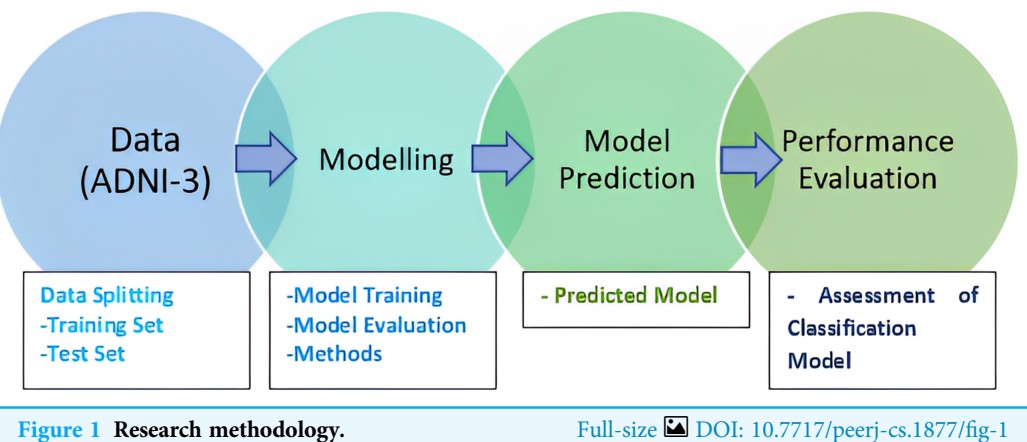

**Figure 1 Research methodology.**

AUC measures the performance of the classification model with a single numerical value. From 0 to 1, the AUC value signifies superior model performance as it approaches higher values. The Matthews correlation coefficient (MCC) is used in binary classification problems and measures the correlation of classification results with actual classes. The MCC is computed on the basis of true positives, true negatives, false positives, and false negatives. The MCC takes a value between −1 and +1. A value of +1 represents excellent forecasting performance, whereas −1 represents completely inverse forecasting performance. A value of 0 represents the same performance as randomly guessing. The MCC is a widely used performance evaluation metric, especially in medical diagnostics, bioinformatics, and genetic research. The metrics are computed using Eqs. (1) to (5).

$$Accuracy = \frac{TP + TN}{TP + TN + FP + FN} \tag{1}$$

$$Precision = \frac{TP}{TP + FP} \tag{2}$$

$$Recall = \frac{TP}{TP + FN} \tag{3}$$

$$F1Score = \frac{2 * Precision * Recall}{Precision + Recall} \tag{4}$$

$$MCC = \frac{TP * TN - FP * FN}{\sqrt{(TP + FN) * (TP + FP) * (TN + FP) * (TN + FN)}} \tag{5}$$

The equations' components include true positive (TP), true negative (TN), false positive (FP), and false negative (FN). The experiments in this study utilized NVIDIA graphics processing unit (GPU) resources using Google Colab. These GPUs provide a considerable advantage in computationally intensive operations such as deep learning. In the study, the results were obtained using the T4 GPU.

## McNemar's test

McNemar's test is widely used, especially in comparing the performance of classification algorithms. It is commonly used when comparing the performance of two classification or diagnostic tests on the same dataset. McNemar's test assesses whether the difference

between these discordant pairs is statistically significant and is used to determine if there is a change or improvement in performance between two tests. It is often used in medical research to compare the effectiveness of different diagnostic methods or treatments (*Bostanci & Bostanci, 2012*). The methodology of the research is presented in Fig. 1.

## RESULTS AND DISCUSSIONS

In this study, a total of 16,766 neuroimaging (MRI) data consisting of three classes (AD, MCI, and CN) were used. Three different CNN models were tested for use in the early diagnosis of AD. Using Python 3.7, the scikit-learn library, and TensorFlow, the models were trained within the Google Colab framework. This platform offers quantitative tools and adaptable resource allocation to manage usage limits and hardware accessibility. In addition, several Python libraries, including Pandas, NumPy, Matplotlib, and Keras, were used to build deep learning models. To work with NIFTI image files, images with the .nii extension were converted to PNG. PNG images were resized to 224 × 224. Images converted to PNG in the dataset were matched to class labels in the CSV file. After the labeling processes were completed, the three selected models were run. Some MRI images of AD, MCI, and CN cases in the dataset are provided in Fig. 2.

A general architecture of the CNN model is presented in Fig. 3. The selected models (EfficientNetB0, DenseNet121, AlexNet) were run for 50 epochs, and the detailed status of the training and test results obtained are given in Tables 3 and 4.

When the training and test results were examined, we found that the AlexNet model had the highest accuracy value during the training. After examining the accuracy values of the test results, we found that the DenseNet121 model performed well. The DenseNet121 model achieved a high MCC value of 0.97. MCC takes a value between 0 and 1 when measuring the performance of a classification model (*Chicco & Jurman, 2020*); the closer the MCC value is to 1, the better the model's classification performance. In the case of DenseNet121, an MCC value of 0.97 indicated that the model had a notably high classification performance, the model's predictions were highly correlated with true classes, and there was a strong correlation between true positives and true negatives. It also highlighted the classification accuracy and reliability of the model. Table 4 shows that the precision value for the DenseNet121 model is 98.19%. The higher the precision value, the lower the probability of false-positive prediction in the model. A higher recall indicates fewer false negative predictions and more true negative predictions. The F1 score is high when both precision and recall are high. When these values are analyzed, the results are promising. The AUC value was computed as 98.22% for the DenseNet121 model; an AUC value close to 1 led us to conclude that the results were good (*Ling, Huang & Zhang, 2003*).

Confusion matrices obtained for all models are given in Fig. 4. According to the confusion matrices in Fig. 4, the DenseNet121 model correctly classified 1,705 of 1,716 AD images, 1,576 of 1,609 MCI images, and 1,658 of 1,705 CN images. In the study by *Savaş (2022)*, 92.98% accuracy was obtained with the EfficientNetB0 model, and 91.91% accuracy was obtained with the EfficientNetB1 model for CN, MCI, and AD classification. In our study, an accuracy of 97.33% was obtained with the EfficientNetB0 model. In the study by

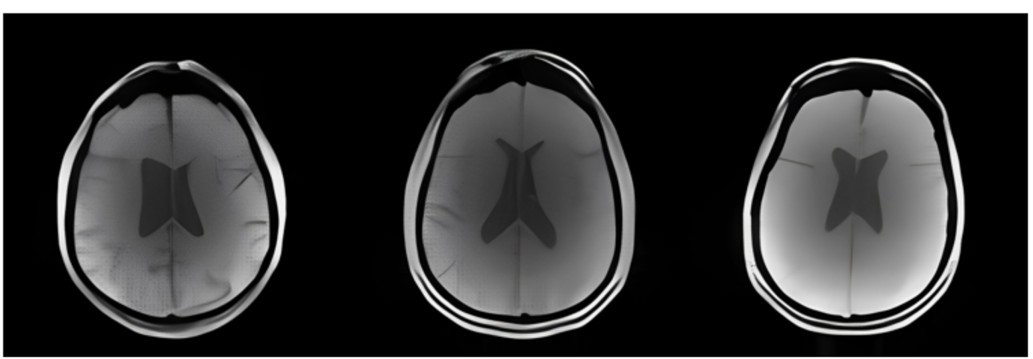

**Figure 2 Examples of AD, MCI, and CN MRI images.**

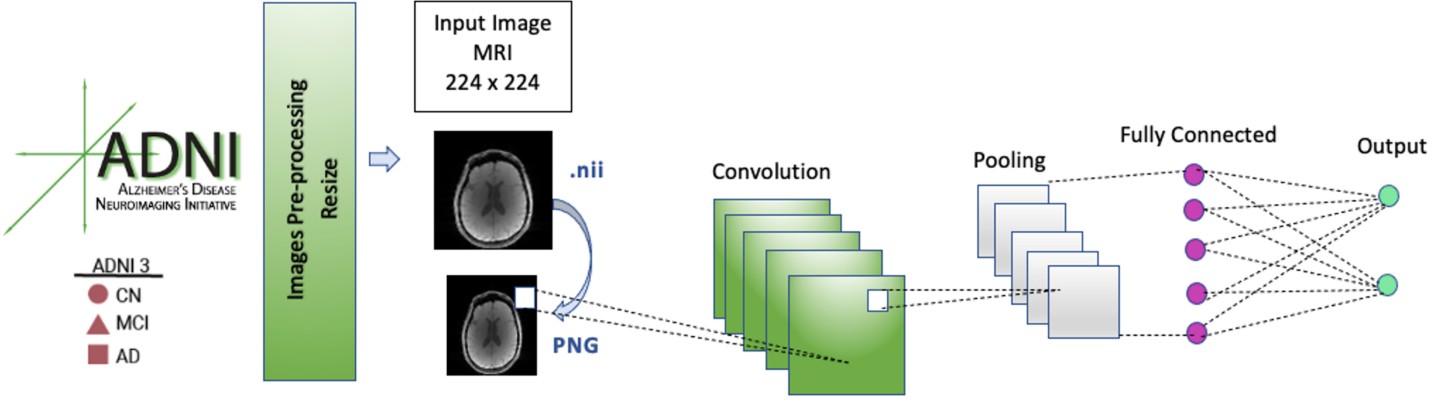

**Figure 3 CNN architecture.**

**Table 3 Training results when models run.**

| Models | Accuracy | Precision | Recall | Auc | F1 score | Validation accuracy |
|---|---|---|---|---|---|---|
| EfficientNetB0 | 0.9920 | 0.9890 | 0.9883 | 0.9994 | 0.9886 | 0.9844 |
| DenseNet121 | 0.9856 | 0.9791 | 0.9778 | 0.9977 | 0.9783 | 0.9858 |
| AlexNet | 0.9977 | 0.9965 | 0.9965 | 0.9996 | 0.9964 | 0.9573 |

**Table 4 Testing results when models run.**

| Models | Accuracy | Precision | Recall | Auc | F1 score | MCC |
|---|---|---|---|---|---|---|
| EfficientNetB0 | 0.9733 | 0.9736 | 0.9729 | 0.9733 | 0.9731 | 0.9599 |
| DenseNet121 | 0.9819 | 0.9819 | 0.9818 | 0.9822 | 0.9819 | 0.9711 |
| AlexNet | 0.9413 | 0.9436 | 0.9398 | 0.9399 | 0.9403 | 0.9106 |

*Khan et al. (2022)*, 95.75% accuracy was obtained with a hybrid model for CN, MCI, and AD classification, whereas the highest accuracy rate was 98.19% with the DenseNet121 model in our study. Thus, our study achieved better results than both previous studies.

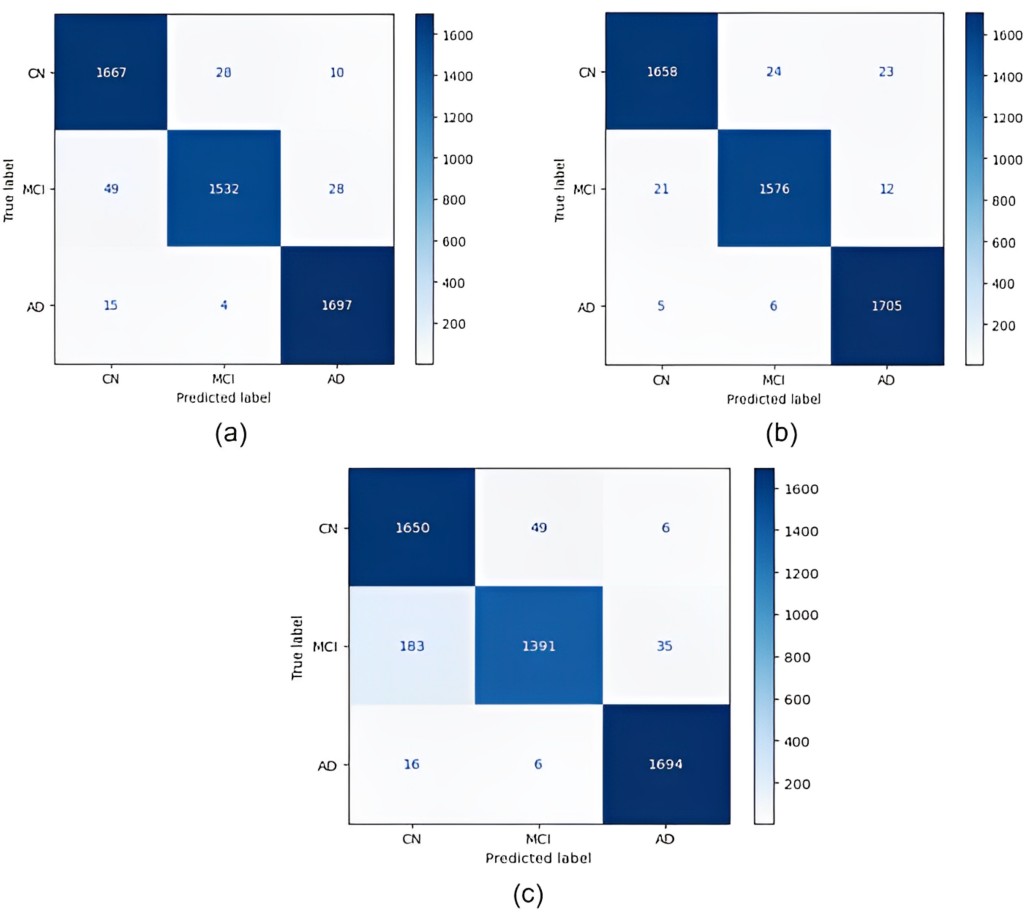

**Figure 4 Confusion matrices.** (A) EfficientNetB0 model (B) DenseNet121 model (C) AlexNet model.

After the images were processed through the models, the models were run using one-*versus*-one classification, in which each class was compared with another class. Table 5 shows the results of the one-*versus*-one classification obtained from the training. Table 6 shows the results of the one-*versus*-one classification obtained from the testing.

The results demonstrated that the EfficientNetB0 model provided satisfactory results in three classifications and high accuracy values for CN *vs* AD and MCI *vs* AD classification. The DenseNet121 model also provided a high accuracy rate of 98.42% for CN *vs* MCI classification. Table 6 shows that the precision value for the EfficientNetB0 model in CN-AD classification is 98.94%, and the AUC value is 98.90%. The table also shows the precision value for the EfficientNetB0 model in the MCI-AD classification, which is 97.95%; the AUC value is 98.00%. In CN-MCI classification, the precision value for the DenseNet121 model is 98.42%, and the AUC value is 98.40%. The higher the precision value, the lower the probability that the model will predict false positives. A higher recall indicates fewer false negative predictions and more true negative predictions. The F1 score

**Table 5 Training results when models run.**

| 1 *vs* 1 | Models | Accuracy | Precision | Recall | Auc | F1 score | Validation accuracy |
|---|---|---|---|---|---|---|---|
| CN/AD | EfficientNetB0 | 0.9990 | 0.9985 | 0.9985 | 0.9999 | 0.9984 | 0.9892 |
| | DenseNet121 | 0.9959 | 0.9939 | 0.9939 | 0.9996 | 0.9938 | 0.9850 |
| | AlexNet | 0.9980 | 0.9970 | 0.9970 | 0.9999 | 0.9970 | 0.9896 |
| MCI/AD | EfficientNetB0 | 0.9987 | 0.9980 | 0.9980 | 1.0000 | 0.9980 | 0.9828 |
| | DenseNet121 | 0.9972 | 0.9958 | 0.9958 | 0.9999 | 0.9957 | 0.9406 |
| | AlexNet | 0.9958 | 0.9937 | 0.9937 | 0.9989 | 0.9937 | 0.9841 |
| CN/MCI | EfficientNetB0 | 0.9975 | 0.9963 | 0.9963 | 0.9996 | 0.9963 | 0.9724 |
| | DenseNet121 | 0.9869 | 0.9869 | 0.9869 | 0.9987 | 0.9968 | 0.9767 |
| | AlexNet | 0.9961 | 0.9942 | 0.9942 | 0.9986 | 0.9942 | 0.9702 |

**Table 6 Testing results when models run.**

| 1 *vs* 1 | Models | Accuracy | Precision | Recall | Auc | F1 score | MCC |
|---|---|---|---|---|---|---|---|
| CN/AD | EfficientNetB0 | 0.9894 | 0.9895 | 0.9895 | 0.9890 | 0.9895 | 0.9790 |
| | DenseNet121 | 0.9716 | 0.9718 | 0.9716 | 0.9720 | 0.9716 | 0.9433 |
| | AlexNet | 0.9774 | 0.9783 | 0.9776 | 0.9780 | 0.9775 | 0.9558 |
| MCI/AD | EfficientNetB0 | 0.9795 | 0.9795 | 0.9800 | 0.9800 | 0.9795 | 0.9594 |
| | DenseNet121 | 0.9075 | 0.9077 | 0.9072 | 0.9070 | 0.9074 | 0.8149 |
| | AlexNet | 0.9750 | 0.9749 | 0.9751 | 0.9750 | 0.9750 | 0.9500 |
| CN/MCI | EfficientNetB0 | 0.9731 | 0.9730 | 0.9734 | 0.9730 | 0.9731 | 0.9664 |
| | DenseNet121 | 0.9842 | 0.9843 | 0.9842 | 0.9840 | 0.9843 | 0.9668 |
| | AlexNet | 0.9655 | 0.9661 | 0.9661 | 0.9660 | 0.9656 | 0.9311 |

is high when both precision and recall are high. According to these values in our results, it can be concluded that satisfactory results were obtained.

Confusion matrices obtained when the models were classified as one-*versus*-one are given in Figs. 5–7.

According to the confusion matrices in Fig. 5, 1,716 to 1,689 AD were classified correctly using the EfficientNetB0 model. The DenseNet121 model classified 1,680 of 1,716 AD patients correctly. The lowest validation rate in AD *vs* CN classification was obtained with the DenseNet121 model.

In the classification of AD *vs* MCI, the EfficientNetB0 model had the best classification results, and the DenseNet121 model had the worst classification results. The complexity matrix of the DenseNet121 model in Fig. 6 shows that 1,575 of 1,716 AD patients were classified correctly.

The most accurate classification of CN *vs* MCI was provided by the DenseNet121 model, according to Fig. 7B. In the complexity matrix of the DenseNet121 model, it is shown that 1,577 of 1,606 MCI classes were correctly classified. The most inaccurate classification was obtained with the AlexNet model.

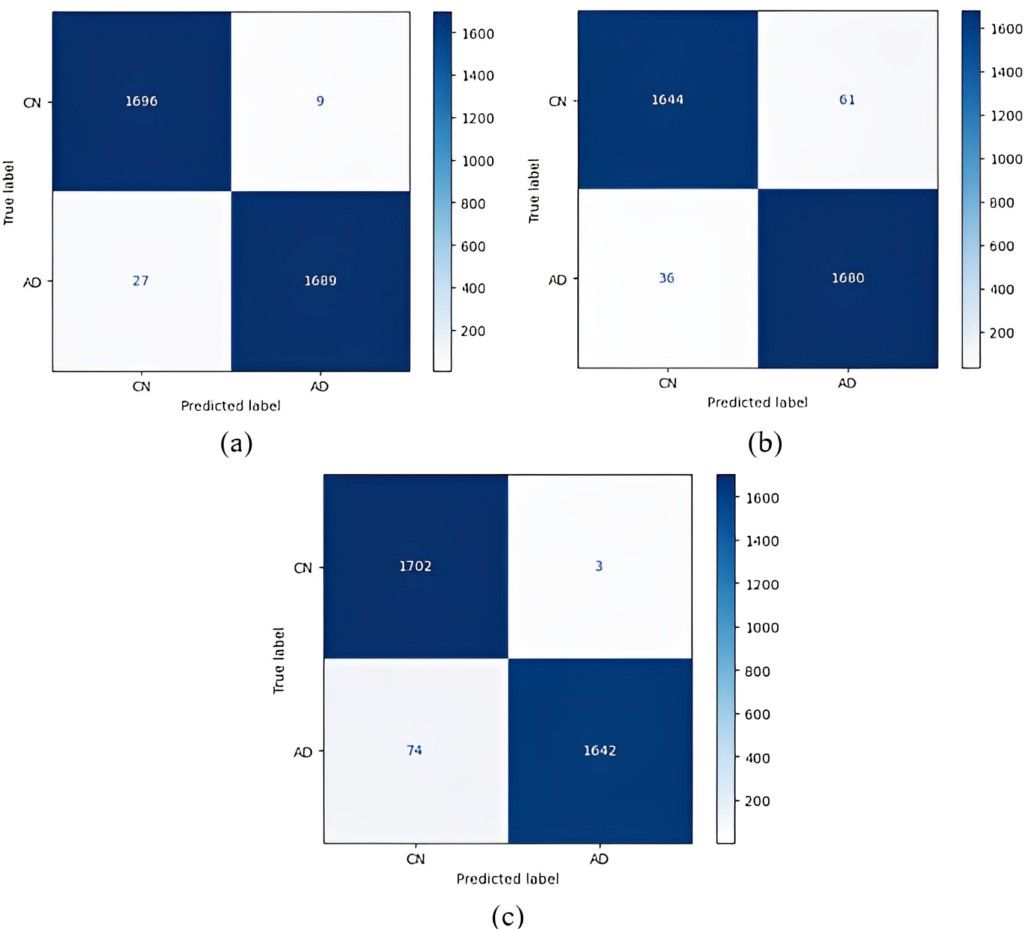

**Figure 5 1 vs 1 classification confusion matrices (CN/AD).** (A) EfficientNetB0 model (B) DenseNet121 model (C) AlexNet model.

After the models were run using the one-*versus*-one classification, they were run using one-*versus*-all classification. In the one-*versus*-all classification, each class formed a group, and all the remaining classes were combined. Table 7 shows the results of the one-*versus*-all classification obtained from the training. Table 8 shows the results of the one-*versus*-all classification obtained from the testing.

The DenseNet121 model for testing achieved excellent accuracy in both MCI *vs* CN-AD and CN *vs* MCI-AD one-*versus*-all classifications. Table 8 shows that the precision value achieved with the DenseNet121 model for MCI-CNAD classification was 98.40%, and the AUC value for this model was 98.00%. Table 8 shows that the precision value of the AlexNet model for AD-CN-MCI classification was 99.58%, and the AUC value was 99.50%. For CN-MCI-AD classification, the precision and AUC values for the DenseNet121 model were 97.83%, and 96.80%, respectively.

The confusion matrices obtained when the models were classified as one-*versus*-all are given in Figs. 8–10.

Figure 8 shows that the DenseNet121 model achieved the best MCI *vs* CN-AD classification performance. With the DenseNet121 model, out of 1,609 CN-AD classes,

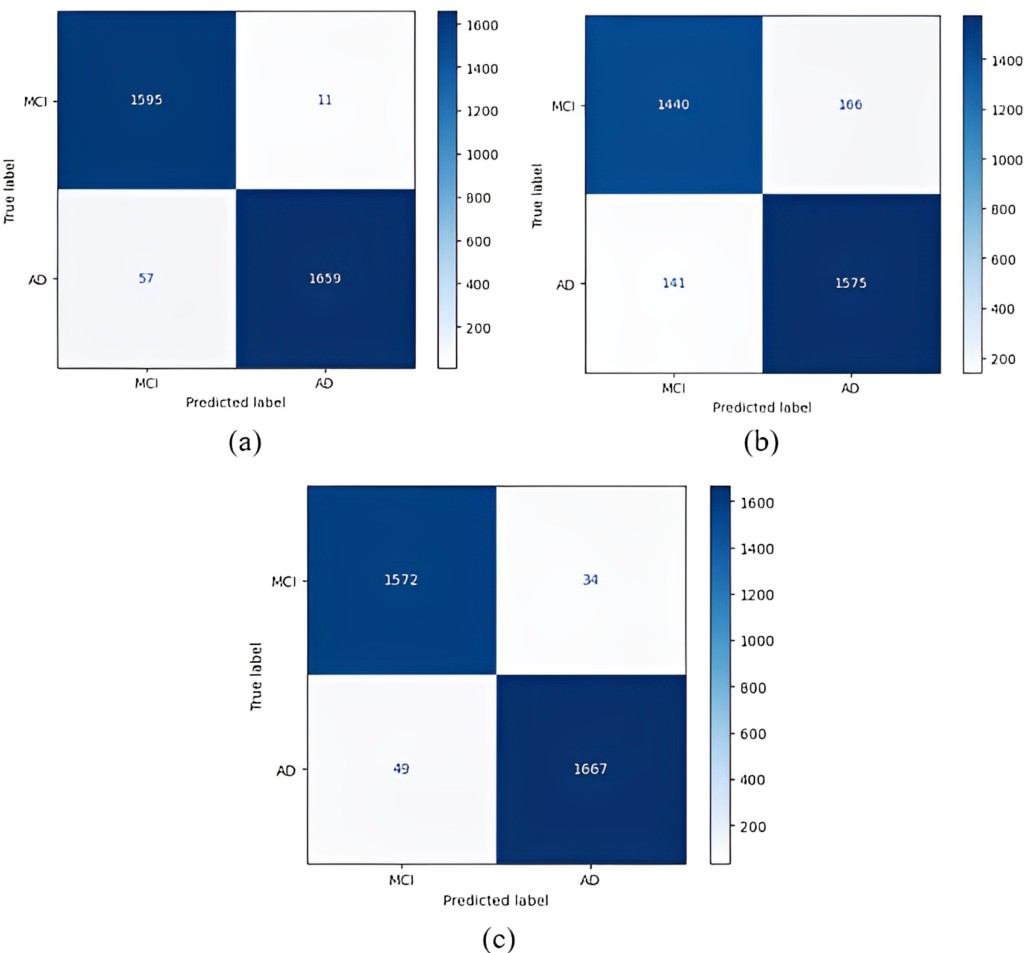

**Figure 6** **1 *vs* 1 classification confusion matrices (MCI/AD).** (A) EfficientNetB0 model (B) Dense-Net121 model (C) AlexNet model. 

1,556 were correctly classified, and for the MCI class, 3,394 of 3,421 classes were classified correctly. On the other hand, the worst classification performance was obtained using the AlexNet model. The AlexNet model correctly classified 1,035 of 1,609 CN-AD classes, and for the MCI class, 3,405 of 3,421 classes were classified correctly.

Figure 9 shows that the AlexNet model achieved the best classification performance in the AD *vs* CN-MCI classification. For the AlexNet model, of 1,716 CN-MCI classes, 1,705 were correctly classified, and 3,304 of 3,314 classes were classified correctly for the AD class. On the other hand, the worst classification performance was obtained using the EfficientNetB0 model. For the EfficientNetB0 model, 1,688 of 1,716 CN-MCI classes were correctly classified, and for the AD class, 3,282 of 3,314 classes were classified correctly.

Figure 10 shows that the DenseNet121 model achieved the best classification performance for CN *vs* MCI-AD classification. For this model, of 3,325 MCI-AD classes, 3,322 were correctly classified, and 1,599 of 1,705 classes were classified correctly for the CN class. On the other hand, the worst classification performance was obtained using the

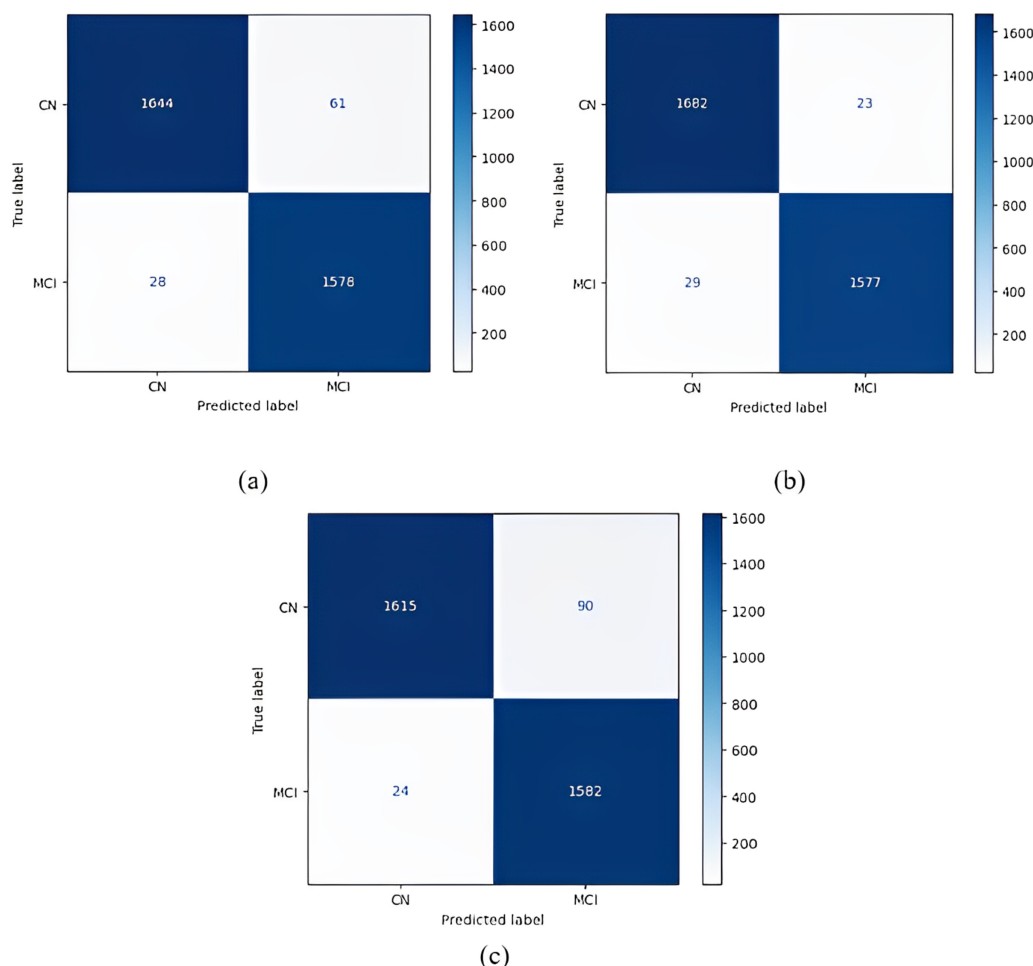

**Figure 7 1 *vs* 1 classification confusion matrices (CN/MCI).** (A) EfficientNetB0 model (B) Dense-Net121 model (C) AlexNet model.

**Table 7 Training results when models run.**

| 1 *vs* All | Models | Accuracy | Precision | Recall | Auc | F1 score | Validation accuracy |
|---|---|---|---|---|---|---|---|
| MCI/CNAD | EfficientNetB0 | 0.9984 | 0.9976 | 0.9976 | 0.9998 | 0.9976 | 0.9790 |
| | DenseNet121 | 0.9970 | 0.9970 | 0.9970 | 0.9999 | 0.9969 | 0.9796 |
| | AlexNet | 0.9969 | 0.9953 | 0.9953 | 0.9993 | 0.9953 | 0.9233 |
| AD/CNMCI | EfficientNetB0 | 0.9929 | 0.9894 | 0.9894 | 0.9991 | 0.9893 | 0.9915 |
| | DenseNet121 | 0.9983 | 0.9983 | 0.9983 | 0.9999 | 0.9982 | 0.9915 |
| | AlexNet | 0.9990 | 0.9985 | 0.9985 | 0.9999 | 0.9985 | 0.9972 |
| CN/MCIAD | EfficientNetB0 | 0.9973 | 0.9959 | 0.9959 | 0.9998 | 0.9959 | 0.9807 |
| | DenseNet121 | 0.9853 | 0.9853 | 0.9853 | 0.9980 | 0.9852 | 0.9702 |
| | AlexNet | 0.9988 | 0.9982 | 0.9982 | 0.9999 | 0.9982 | 0.9776 |

**Table 8 Testing results when models run.**

| 1 *vs* all | Models | Accuracy | Precision | Recall | Auc | F1 score | MCC |
|---|---|---|---|---|---|---|---|
| MCI/CNAD | EfficientNetB0 | 0.9705 | 0.9649 | 0.9677 | 0.9680 | 0.9663 | 0.9326 |
| | DenseNet121 | 0.9840 | 0.9838 | 0.9796 | 0.9800 | 0.9816 | 0.9633 |
| | AlexNet | 0.8827 | 0.9203 | 0.8193 | 0.8190 | 0.8492 | 0.7362 |
| AD/CNMCI | EfficientNetB0 | 0.9880 | 0.9865 | 0.9870 | 0.9870 | 0.9867 | 0.9734 |
| | DenseNet121 | 0.9948 | 0.9934 | 0.9951 | 0.9950 | 0.9943 | 0.9885 |
| | AlexNet | 0.9958 | 0.9954 | 0.9953 | 0.9950 | 0.9954 | 0.9874 |
| CN/MCIAD | EfficientNetB0 | 0.9757 | 0.9733 | 0.9725 | 0.9730 | 0.9729 | 0.9458 |
| | DenseNet121 | 0.9783 | 0.9836 | 0.9685 | 0.9680 | 0.9754 | 0.9665 |
| | AlexNet | 0.9654 | 0.9738 | 0.9500 | 0.9500 | 0.9605 | 0.9234 |

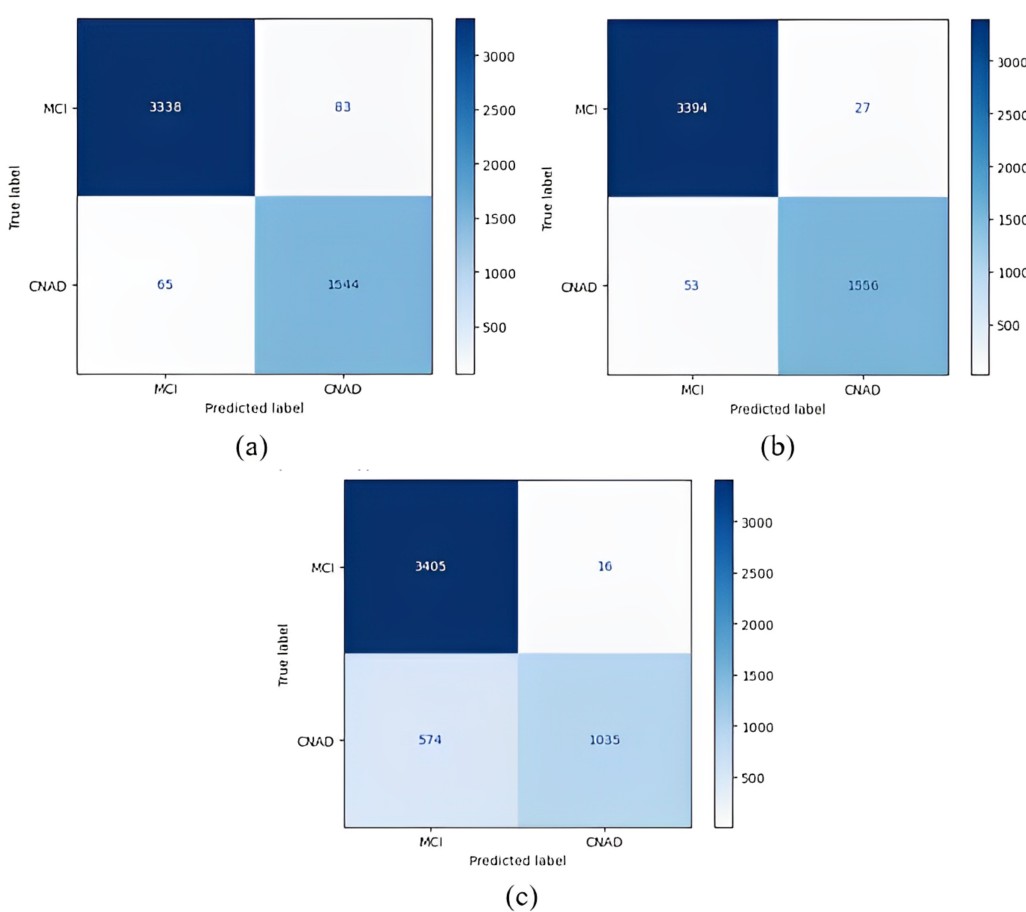

**Figure 8 1 *vs* All classification confusion matrices (MCI/CNAD).** (A) EfficientNetB0 model (B) DenseNet121 model (C) AlexNet model.

AlexNet model. For this model, in CN *vs* MCI-AD classification, 3,318 of 3,325 MCI-AD classes were correctly classified, and for the CN class, only 1,538 of 1,705 classes were classified correctly.

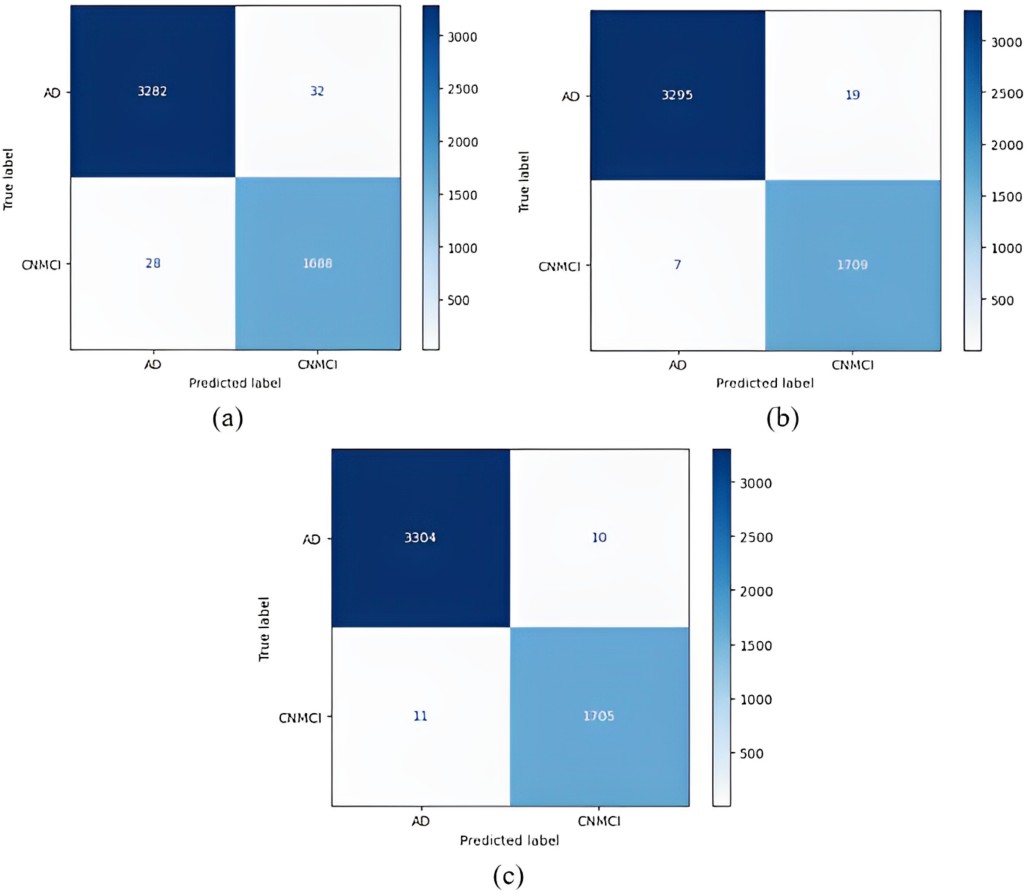

**Figure 9 1 *vs* All classification confusion matrices (AD/CNMCI).** (A) EfficientNetB0 model (B) DenseNet121 model (C) AlexNet model.               

When using one-*versus*-one classification, the EfficientNetB0 model provided the best results, with 98.94% accuracy for CN *vs* AD classification. The EfficientNetB0 model also provided the best results for MCI *vs* AD classification, with 97.95% accuracy. Finally, the DenseNet121 model provided the best results for CN *vs* MCI classification, with 98.42% accuracy. Through analyzing these results, we determined that these high accuracy values are different because each classification task has different dynamics, and the data distribution of each class is different. AD *vs* MCI classification accuracy is lower than the others because the distinction between an individual with AD and an individual with a mild stage of this disease is more complex.

In this study, McNemar's test was used as the nonparametric statistical variant of the $\chi 2$ test to determine the statistical significance between the performances of the classifiers. When comparing the performances of two classifiers, four possible outputs exist. These outputs can be seen in Table 9.

As shown in Table 9, $N_{ff}$, $N_{sf}$, $N_{fs}$, and $N_{ss}$ represent the number of times both classifiers failed to predict, only classifier A succeeded, only classifier B succeeded, and both classifiers succeeded, respectively. However, only $N_{sf}$ and $N_{fs}$ values were used to obtain a significant difference because these values represented the number of times a classifier

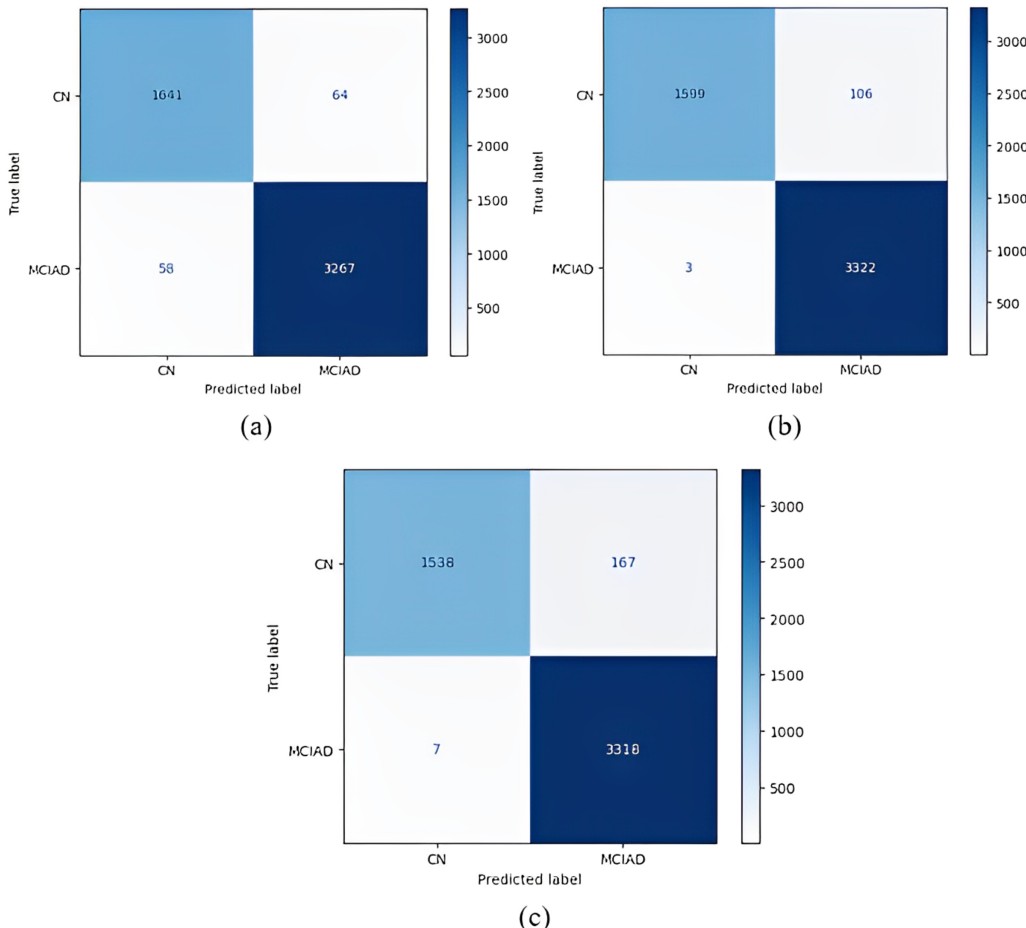

**Figure 10 1 *vs* all classification confusion matrices (CN/MCIAD).** (A) EfficientNetB0 model (B) DenseNet121 model (C) AlexNet model.

**Table 9 Possible outcomes of two classifiers.**

|  | Classifier A failed | Classifier A succeeded |
|---|---|---|
| Classifier B failed | $N_{ff}$ | $N_{sf}$ |
| Classifier B succeeded | $N_{fs}$ | $N_{ss}$ |

succeeded and another failed. $N_{sf}$ and $N_{fs}$ values were employed to calculate the z-score, which represents whether two classifiers show similar performance, as shown in Eq. (6):

$$Z = \frac{\left(\left|N_{sf} - N_{fs}\right| - 1\right)}{\sqrt{N_{sf} + N_{fs}}} \tag{6}$$

If the z-score equals 0, this situation is accepted because the two classifiers show similar performance. As the z-score diverges from 0, the performance difference of the two classifiers becomes more significant. In addition, z-scores can be interpreted according to

**Table 10 Z scores and confidence levels.**

| z score | One-tailed prediction | Two-tailed prediction |
| --- | --- | --- |
| 1.345 | 95% | 90% |
| 1.960 | 97.5% | 95% |
| 2.326 | 99% | 98% |
| 2.576 | 99.5% | 99% |

**Table 11 Z scores of architectures on Alzheimer's disease prediction.**

| 1 *vs* 1 | CN *vs* AD | | |
| --- | --- | --- | --- |
| | EfficientNetB0 | DenseNet121 | AlexNet |
| EfficientNetB0 | – | ← 5.86 | ←4.44 |
| DenseNet121 | – | – | ↑1.57 |
| AlexNet | – | – | – |
| | CN *vs* MCI | | |
| | EfficientNetB0 | DenseNet121 | AlexNet |
| EfficientNetB0 | – | ↑ 3.66 | ←1.94 |
| DenseNet121 | – | – | ←5.23 |
| AlexNet | – | – | – |
| | MCI *vs* AD | | |
| | EfficientNetB0 | DenseNet121 | AlexNet |
| EfficientNetB0 | – | ←13.00 | ←1.31 |
| DenseNet121 | – | – | ↑11.99 |
| AlexNet | – | – | – |

the confidence levels for one-tailed and two-tailed predictions. In Table 10, confidence levels corresponding to the z-scores are presented.

The z-scores of the architectures for AD prediction are given in Tables 11 and 12.

Tables 11 and 12 compare the performance of different classifiers on the given dataset, using arrowheads (←, ↑) to indicate which classifier performed better in terms of true predictions (both true positives and true negatives). The z-scores next to the arrowheads measure how statistically significant the results are. Furthermore, statistically significant results are written in bold in the tables. If the results are statistically significant, the confidence levels are included below for both one-tailed and two-tailed predictions.

Table 11 represents the one-*versus*-one classification results of the three deep learning architectures for CN *vs* AD, CN *vs* MCI, and MCI *vs* AD classification. The results of CN *vs* AD classification shown in Table 11 demonstrate that EfficientNetB0 performed better than DenseNet121 and AlexNet. In addition, the performance differences between EfficientNetB0 and the other architectures were statistically significant for CN and AD classification. Z-score values of 5.86 and 4.44 indicate 99.5% and 99% confidence levels.

**Table 12 Z scores of architectures on Alzheimer's disease prediction.**

| 1 *vs* all | CN *vs* MCIAD | | |
| --- | --- | --- | --- |
| | EfficientNetB0 | DenseNet121 | AlexNet |
| EfficientNetB0 | – | ↑0.86 | ←3.41 |
| DenseNet121 | – | – | ←5.87 |
| AlexNet | – | – | – |
| | **MCI *vs* CNAD** | | |
| | EfficientNetB0 | DenseNet121 | AlexNet |
| EfficientNetB0 | – | ↑4.94 | ←17.46 |
| DenseNet121 | – | – | ←21.36 |
| AlexNet | – | – | – |
| | **AD *vs* CNMCI** | | |
| | EfficientNetB0 | DenseNet121 | AlexNet |
| EfficientNetB0 | – | ↑3.84 | ↑4.87 |
| DenseNet121 | – | – | ↑0.70 |
| AlexNet | – | – | – |

AlexNet demonstrated higher performance than DenseNet121, and this performance difference is statistically significant, with confidence levels of 95% and 90%.

When we analyzed the results of CN *vs* MCI, as shown in Table 11, we concluded that DenseNet121 performed better than both EfficientNetB0 and AlexNet. In addition, the performance differences between DenseNet121 and the other architectures were statistically significant for CN and MCI classification. Z-score values of 3.66 and 5.23 indicate 99.5% and 99% confidence levels. EfficientNetB0 demonstrated higher performance than AlexNet, and this performance difference is statistically significant, with confidence levels of 95% and 90%.

After examining the results of MCI *vs* AD classification, as shown in Table 11, we concluded that EfficientNetB0 performed better than both DenseNet121 and AlexNet. However, the performance difference between EfficientNetB0 and AlexNet was statistically significant with a z-score of 1.31. In contrast, the performance difference between EfficientNetB0 and DenseNet121 was statistically significant, with confidence levels of 99.5% and 99% for MCI and AD classification. AlexNet demonstrated higher performance than DenseNet121, and this performance difference is statistically significant, with confidence levels of 99.5% and 99%.

As shown in Table 12, the results of CN *vs* MCI-AD classification demonstrate that DenseNet121 outperformed the other architectures. The performance difference between DenseNet121 and EfficientNetB0 was not found to be statistically significant, with a z-score of 0.86, whereas the performance difference between DenseNet121 and AlexNet was found to be statistically significant, with confidence levels of 99.5% and 99% for CN and other (MCI-AD) classification. Furthermore, EfficientNetB0 outperformed AlexNet, and

the performance difference was statistically significant, with confidence levels of 99.5% and 99%.

The results of MCI *vs* CN-AD classification, as shown in Table 12, demonstrate that DenseNet121 outperformed the other architectures. The performance differences between DenseNet121 and the other architectures were statistically significant, with confidence levels of 99.5% and 99% for MCI and other (CN-AD) classifications. Furthermore, EfficientNetB0 outperformed AlexNet, and the performance difference was found to be statistically significant, with confidence levels of 99.5% and 99%.

In addition, the results shown in Table 12 demonstrate that AlexNet outperformed the other architectures in AD *vs* CNCMI classification. The performance difference between AlexNet and DenseNet121 was not statistically significant, with a z-score of 0.70 for AD and other (CN-MCI) classifications. However, the performance difference between AlexNet and EfficientNetB0 was found to be statistically significant, with confidence levels of 99.5% and 99%. Furthermore, DenseNet121 outperformed EfficientNetB0, and the performance difference was statistically significant, with confidence levels of 99.5% and 99%.

*Mora-Rubio et al. (2023)* obtained an accuracy of 89.02% with the vision transformer (ViT) in AD classification *vs* CN. In our study, the EfficientNetB0 model achieved an accuracy of 98.94% for CN *vs* AD classification. At the same time, *Mora-Rubio et al. (2023)* obtained an accuracy of 66.41% with DenseNet and EfficientNet models for MCI *vs* CN classification. Our result was an accuracy of 98.42% obtained with the DenseNet121 model for MCI *vs* CN classification. We obtained better results in one-*versus*-one classification, which may be due to the differences in the numbers in the dataset, model architectures used, and testing and training processes. When our results are compared with other studies in the literature (*Savaş, 2022*), the highest average accuracy rate of 92.98% was achieved using the EfficientNetB0 model, whereas *Khan et al. (2022)* reported a maximum average accuracy rate of 95.75%, which was obtained using an XGB + DT + SVM hybrid model.

Compared with these results, our average accuracy rate with the DenseNet121 model was 98.19%. The EfficientNetB0 model follows, with an average accuracy rate of 97.33%. Our results show better performance compared with those of previous studies. Our choice of preferred models compared with the models used in the study by *Khan et al. (2022)* allowed us to obtain better results. At the same time, the selection of more recent models might have also affected our results. When examining the results of other studies, it can be observed that in the work of *Mehmood et al. (2021)*, a 98.73% accuracy rate was achieved for the classification of CN *vs* AD. In addition, *Sethi et al. (2022)* reported an accuracy rate of 89.40% for AD *vs* CN classification.

Furthermore, *Naz, Ashraf & Zaib (2021)* obtained an accuracy of 98.89% using the VGG-16 model and 91.38% using the AlexNet model in AD *vs* CN classification. These findings indicate that our study produced better results in the classification of AD *vs* CN compared with the results reported in the literature. In addition, the EfficientNetB0 model achieved a better result in CN *vs* AD classification, with an accuracy rate of 98.94%.

Regarding MCI *vs* AD classification, *Naz, Ashraf & Zaib (2021)* obtained a slightly higher accuracy result of 99.27% using the VGG-19 model. They obtained an accuracy of

97.06% in the classification of CN *vs* MCI; in our study, an accuracy of 98.42% was obtained with the EfficientNet121 model. Because it is important to distinguish between a CN subject and a subject with MCI, the high classification accuracy of CN *vs* MCI marks an important contribution. In addition, studies in the literature using one-*versus*-all classification are limited for the related problem. Promising results have been obtained for this type of classification in our study.

In the study by *Islam & Zhang (2017)*, the OASIS dataset was used. They performed the classification with a total of four classes and 416 subjects. By proposing their model framework, the best result of 73.75% was achieved in 10 epochs using five-fold cross-validation. In our study, there were three classes in total conducted on the ADNI dataset. After necessary operations were performed on the 16,766 MRI images in the dataset, the three models were run by separating the dataset into 70% for training and 30% for testing. When examining the results, our approach outperformed that of the study by *Islam & Zhang (2017)*. More MRI images were used in our study than in theirs. Larger datasets have the potential to reduce a model's tendency to memorize; when a model is trained with more data, it tends to learn general patterns and features rather than memorizing situations during the learning process.

In a study conducted by *Mamun et al. (2022)*, 6,219 MRI images with four classes were used. The dataset was first preprocessed by performing image resizing, noise removal, image segmentation, and smoothing. They trained the dataset using the holdout cross-validation method and obtained results using a CNN architecture. The highest accuracy of 97.60% was achieved with their approach. In our study, a general average accuracy rate of 98.19% was obtained using the DenseNet121 model. The one-*versus*-one and one-*versus*-all classification accuracies were higher than the results obtained in this study. Another difference between our study and the study by *Mamun et al. (2022)* is that we used more MRI images with three classes, and the dataset was divided into 70% for training and 30% for testing when running the models. In addition, the EfficientNetB0, DenseNet121, and AlexNet models were employed to perform the one-*versus*-one and one-*versus*-all classifications.

*Sekhar & Jagadev (2023)* used a dataset of 6,400 MRI images for the classification of AD by dividing the dataset into training, testing, and validation portions. They proposed their own neural network. They also stated that the advantage of end-to-end learning is that it can create an effective visual explanation of the logic behind the classification results obtained by CNN, which will help doctors understand the impact of CNN and find new biomarkers. The best result of 98.5% was obtained with the EfficientNet model. In our study, an accuracy of 98.94% was achieved with the EfficientNetB0 model in CN-AD classification for one-*versus*-one classification. Furthermore, an accuracy of 99.58% was obtained using the AlexNet model in AD-CN-MCI one-*versus*-all classification. This shows that considerable improvements have been made in our study compared with the results of others in the literature.

The results of our study clearly demonstrate that the DenseNet121 model provides the worst results for one-*versus*-one classification (MCI *vs* AD), with an accuracy of 90.75%. AD diagnosed at an early stage can improve patients' quality of life and slow the

progression of the disease. Our study obtained the best CN *vs* MCI and CN *vs* AD classification results. Because it is important to distinguish between CN and MCI, the accuracy rate obtained in this study is a significant contribution to the literature. Early detection of the disease in the mild stage is also an important tool for preventing the progression of AD. When it comes to distinguishing between a patient with AD and a patient with MCI, determining which class they belong to becomes more challenging. In contrast, for less complex cases, better results were achieved, and the classification process was relatively easier.

## CONCLUSION AND FUTURE WORK

This study aims to enable early detection and classification of AD using CNN-based methods. The ADNI-3 dataset was utilized, which consisted of T1-weighted structural MRI data and axial brain slices from 515 individuals.

Within the scope of this study, the images in NIFTI format within the dataset were converted to PNG format as the preliminary step. Each converted image was resized to a dimension of $224 \times 224$. The PNG images were paired with their corresponding class labels in the CSV file. The dataset was split into 70% for training and 30% for testing, and results were obtained with the models determined in the next stages. Results were obtained from the models using both one-*versus*-one classification and one-*versus*-all classification.

Satisfactory results were obtained for CN *vs* MCI one-*versus*-one classification. A significant finding was the ability to distinguish a healthy person from someone with MCI. Early detection of AD, before progression in MCI individuals, is a major contribution of this study. A one-*versus*-one classification can be an important tool in early diagnosis, management, treatment, and research on AD. Such classifications can potentially reduce the disease's effects and improve patients' quality of life. In addition, some important factors must be considered before the models are clinically applicable. One of these is that the model should provide a high accuracy rate and produce reliable results. In some clinical settings, misleading results, erroneous diagnoses, and predictions are unacceptable. Likewise, in some clinical applications, medical experts can analyze MRI images to determine the AD status of patients. By examining the results of the model, medical experts can see possible signs of AD and then support their diagnosis by initiating the necessary medical procedures. The results of the model can be combined with the medical experts' clinical experience and patient history and then used to assist in making the final diagnosis. Therefore, we believe that these models can be of significant benefit in the clinical setting.

McNemar's test is an important tool for comparing and evaluating the accuracy of classification models. A nonparametric statistical variant of the $\chi 2$ test, McNemar's test, was used to determine the statistical significance between the performances of the classifiers. McNemar's test is useful at this point because it is important to evaluate whether the difference between the models is statistically significant, enabling researchers to choose the right model and make decisions for development. The overall results obtained for the early detection of AD are promising, and it is anticipated that they could be adapted to a

more general approach with different test datasets. Although the model's results are satisfactory, expanding the dataset might lead to even better outcomes.

In future research, we plan to focus on augmenting the dataset using generative adversarial networks (GAN) to generate artificial MRI images. The potential benefits of using GAN-generated images in the study will be investigated.

### Funding
The authors declare that they have no competing interests.

### Competing Interests
The authors received no funding for this work.

### Author Contributions
- Begüm Şener conceived and designed the experiments, performed the experiments, analyzed the data, performed the computation work, prepared figures and/or tables, authored or reviewed drafts of the article, and approved the final draft.
- Koray Acici analyzed the data, prepared figures and/or tables, authored or reviewed drafts of the article, and approved the final draft.
- Emre Sümer conceived and designed the experiments, analyzed the data, performed the computation work, authored or reviewed drafts of the article, and approved the final draft.

### Data Availability
The data is available from ADNI (https://adni.loni.usc.edu).

The data is available at GitHub and Zenodo:

- https://github.com/Begumer/Alzheimer-s-Disease-Classification/tree/main.

- Begumer. (2023). Begumer/Alzheimer-s-Disease-Classification: Alzheimer's (Alzheimer's). Zenodo. https://doi.org/10.5281/zenodo.10259112.

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
