# Peer review of "Categorization of Alzheimer’s disease stages using deep learning approaches with McNemar’s test"

_PeerJ Computer Science, doi:10.7717/peerj-cs.1877_

## Round 0.1 · original submission · Major Revisions

Hi,

Kindly address all the comments from expert reviewers appropriately.

·

Basic reporting

The manuscript's strength lies in its clear objective and the application of deep learning to Alzheimer's disease detection. However, it could be enhanced by providing a more comprehensive literature review, particularly comparing the proposed methods with recent advancements in the field. For instance, a study by Islam and Zhang (2017) demonstrated a novel deep learning model for Alzheimer’s Disease detection using Brain MRI Data, which could serve as a benchmark for comparison. Additionally, the manuscript should ensure data availability in line with PeerJ's standards for reproducibility, as emphasized by Mamun et al. (2022), who provided datasets and code for replication.

Suggested Improvements:

Include a more detailed comparison with recent studies such as Islam and Zhang (2017) and Mamun et al. (2022).
Ensure the availability of datasets and code, or provide a clear explanation for any restrictions.

Experimental design

While the experimental design is robust, it could be improved by detailing the rationale behind the choice of deep learning models and parameters. The study by Sekhar and Jana (2023) implemented Alzheimer’s disease detection using U-Net and EfficientNet, discussing the choice and tuning of these models. Moreover, the manuscript should include a comprehensive statistical analysis, as performed by Mamun et al. (2022), to support the validity of the findings.

Suggested Improvements:

Elaborate on the selection and optimization of deep learning models, drawing comparisons with studies like Sekhar and Jana (2023).
Provide a detailed statistical analysis, including assumptions and robustness checks.

Validity of the findings

The findings are promising, but the manuscript should address the potential for overfitting and the model's real-world applicability. Comparative analysis with other models, such as the one by Mamun et al. (2022), which achieved an accuracy of 97.60%, would contextualize the results. Additionally, the manuscript could benefit from discussing the generalizability of the findings, as suggested by Sekhar and Jana (2023), who emphasized the importance of model performance in clinical settings.

Suggested Improvements:

Discuss the generalizability and clinical applicability of the models, referencing studies like Sekhar and Jana (2023).
Include a comparative analysis with other recent models, such as those discussed by Mamun et al. (2022).

Additional comments

The manuscript is a valuable addition to the field, yet it would benefit from a discussion on ethical considerations and broader implications for healthcare integration, as highlighted by Mamun et al. (2022). The potential impact on patient outcomes and healthcare systems should be considered, drawing insights from the ensemble deep learning framework proposed by Sekhar and Jana (2023), which emphasized low parameter count and better training time, important for real-world applications.

In conclusion, while the manuscript presents significant contributions, addressing these points would align it more closely with PeerJ's standards and the current state of research in the field.

Reviewer 2 ·

Basic reporting

The article demonstrates good potential but necessitates significant revisions. The figures and tables within the manuscript have been meticulously examined; however, the code provided in the Github repository's notebooks lacks reproducibility.

Experimental design

While the article adeptly elucidates the utilization of three deep learning ML models, it falls short in justifying their application specifically for Alzheimer's Disease (AD) prediction. For instance, the rationale behind not employing ML models akin to XGBoost or LightGBM remains unexplored. Additionally, while McNemar’s test is widely recognized, employing methods such as combinatorial fusion analysis could have offered a more appropriate means to compare the performance of ML models.

Validity of the findings

Further insight into the identified targets for AD, as outlined in (https://www.nature.com/articles/s42003-022-03068-7), would have strengthened the article. Line 217-219 requires a paraphrase regarding the data source, while specifying the proportions in each set (testing and training/validation) at Line 256 is crucial. Multiple sections throughout the article necessitate language refinement, prompting the authors to conduct additional checks for clarity and coherence.

---

## Round 0.2 · accepted · Accept

Congratulations on the acceptance of your manuscript.

Reviewer 2 ·

Basic reporting

Authors have addressed the points raised in the previous review well. No further review of the article is required and the article is suitable for acceptance.

Experimental design

NA

Validity of the findings

NA